# Zero-Sum Positional Differential Games as a Framework for Robust Reinforcement Learning: Deep Q-Learning Approach

## Abstract

Robust Reinforcement Learning (RRL) is a promising Reinforcement Learning (RL) paradigm aimed at training robust to uncertainty or disturbances models, making them more efficient for real-world applications. Following this paradigm, uncertainty or disturbances are interpreted as actions of a second adversarial agent, and thus, the problem is reduced to seeking the agents' policies robust to any opponent's actions. This paper is the first to propose considering the RRL problems within the positional differential game theory, which helps us to obtain theoretically justified intuition to develop a centralized Q-learning approach. Namely, we prove that under Isaacs's condition (sufficiently general for real-world dynamical systems), the same Q-function can be utilized as an approximate solution of both minimax and maximin Bellman equations, and we also indicate conditions when this Q-function can be decomposed. Based on these results, we present the Isaacs Deep Q-Networks (IDQN) and Decomposed Isaacs Deep Q-Networks (DIDQN) algorithms, respectively. We analyze their performance by comparing them with other baseline RRL and Multi-Agent RL algorithms. We consider both simple environments with known accurate solutions and complex large-dimensional MuJoCo environments. In each experiment, we thoroughly evaluate the agents' policies obtained after learning, training opponents against them using various RL algorithms with various parameters. The experiment results demonstrate the superiority of the presented algorithms in all experiments under consideration.

## 1 Introduction

In the last ten years, neural network models trained by Reinforcement Learning (RL) algorithms (Sutton & Barto (2018)) have demonstrated outstanding performance in various game and physics simulators (see, e.g., Mnih et al. (2015); Silver et al. (2017); OpenAI (2018); Vinyals et al. (2019); Liu et al. (2022)). However, the usage of such models for practical problems is still limited due to their instability to uncertainty or disturbances occurring in the real world. One promising paradigm to overcome these difficulties is Robust Reinforcement Learning (RRL) Morimoto & Doya (2000) (see also Robust Adversarial RL Pinto et al. (2017)), in which such uncertainty or disturbances are interpreted as actions of a second adversarial agent, and thus the problem is reduced to seeking the agents' policies robust to any opponent's actions.

The fundamental difficulty of RRL problems, as a particular case of Multi-Agent RL (MARL) problems, is the non-stationary of the environment (see, e.g., Busoniu et al. (2008); Zhang et al. (2021)) from the point of each agent's view. Often, this leads to the failure (Lanctot et al. (2017)) of the decentralized (independent) learning and the design of a centralized approach allowing agents to exchange information during the learning (see, e.g., Lowe et al. (2017); Foerster et al. (2018)). The exchange can utilize a shared memory, a shared policy, but more often, a shared Q-function, which requires an adequate theory within which such a function exists.

According to the Markov game theory (see, e.g., Shapley (1953); Bertsekas (1976); Van Der Wal (1980)), any zero-sum game has a value (Nash equilibrium) which can be used to construct such shared Q-function (Littman (1994)). However, even in the simplest examples of Markov games (e.g., paper-rock-scissors), optimal policies may not be pure (deterministic) and, therefore, can pro-

vide only the expected value of a payoff. Thus, the Markov game theory may be inappropriate if, according to the problem statement (for example, in the case of developing safe control systems), it is required to seek robust policies guaranteeing a deterministic payoff value.

In this paper, we make the following contributions to the RRL research:

- We are the first to propose considering the RRL paradigm within the framework of the positional differential game theory (Krasovskii & Subbotin (1987); Subbotin (1995)), which makes it possible to study pure agents' policies and deterministic payoff values.

- We prove that under Isaacs's condition (Isaacs (1965)) (sufficiently general and easily verifiable for real-world dynamical systems), the same Q-function can be utilized as an approximate solution of both minimax and maximin Bellman equations. We also indicate a condition when this Q-function can be decomposed (Sunehag et al. (2017)). Thus, we present a theoretically justified intuition for developing a centralized Q-learning approach.

- Taking this intuition into account, we present the Isaacs Deep Q-Networks (IDQN) and Decomposed Isaacs Deep Q-Networks (DIDQN) algorithms as natural extensions of the single-agent DQN algorithm (Mnih et al. (2015)). The experiment results demonstrate the superiority of the presented algorithms compared to RRL and MARL baselines (see Fig. 3).

- We offer to test RRL algorithms on new environments originating from differential game examples with known accurate solutions. Such environments can serve as additional reliable tests in future research of RRL algorithms.

- We consider a framework for thorough evaluating the robustness of trained policies based on using various RL algorithms with various parameters (see Fig. 2). We hope this framework will become the new standard in research of continuous RRL problems as well as continuous MARL problems in zero-sum setting.

## 1.1 RELATED WORK

In recent years, the RRL paradigm has been successfully used as an effective tool for finding a policy robust to various environment's physical parameters (such as mass, friction, etc.) (see, e.g., Pinto et al. (2017); Abdullah et al. (2019); Zhai et al. (2022)). This paper does not study such properties of policies, focusing only on their robustness to dynamical uncertainty or disturbances interpreted as an adversary agent.

As mentioned above, RRL problems can be naturally considered as non-cooperative Markov games (Pinto et al. (2017)) and solved by the corresponding algorithms of decentralized (see Tampuu et al. (2017); Gleave et al. (2019)) or centralized (see Lowe et al. (2017); Li et al. (2019); Kamalaruban et al. (2020)) learning. We consider some of these algorithms as a baselines for the comparative analysis in our experiments (see Experiments).

DQN extension to zero-sum Markov games was carried out in Fan et al. (2020); Zhu & Zhao (2020); Phillips (2021); Ding et al. (2022), where the main idea was to solve minimax Bellman equations in the class of mixed policies. The presented in this paper IDQN and DIDQN algorithms of solving zero-sum differential games seek solutions of such equations in pure policies, significantly reducing running time and improving performance (see the comparison with NashDQN in Experiments).

The first formalization of RRL problems within the framework of differential games was proposed in fundamental paper Morimoto & Doya (2000), where a particular class of games called $H_\infty$-control (Başar & Bernhard (1995); Zhou et al. (1996)) was considered. This approach was further developed in Al-Tamimi et al. (2007); Han et al. (2019); Zhai et al. (2022). We do not consider the algorithms from these papers in our experiments since the $H_\infty$-control theory represents another perspective on optimality compared to the classical differential game theory (Isaacs (1965)). Nevertheless, we note that Al-Tamimi et al. (2007) established the existence of shared Q-functions for linear differential games, which can be interpreted as a particular case of our theoretical results (Theorem 1).

Applying RL algorithms to solve pursuit-evasion differential games within the classical theory was studied in Wang et al. (2019); Jiang et al. (2020); Xu et al. (2022); Selvakumara & Bakolas (2022). However, such a class of games seems the most complex for directly applying RL since the agent has too little chance of reaching the aim (capture) in the exploration stage and may not have enough

informative samples for further learning. To overcome these difficulties, these papers suggest modifying reward functions, which naturally increase the number of informative samples but, in fact, change the problem statement. Finite-horizon differential games considered in this paper do not have uninformative samples and seem more suitable for applying RL (Harmon et al. (1996)).

The closest study to our paper is Li et al. (2022) developing DQN for solving infinite-horizon reach-avoid zero-sum differential games. The fundamental difference of the algorithm proposed in Li et al. (2022) is that the second player knows the first agent's next action in advance. Note that this approach does not require a shared Q-function, and therefore we also test it in our experiments (see CounterDQN) to assess the significance of the shared Q-function usage for performance.

There are extensive studies (see, e.g., Patsko (1996); Bardi & Dolcetta (1997); Cardaliaguet et al. (1999); Kumkov et al. (2005); Kamneva (2019); Lukoyanov & Gomoyunov (2019)) on numerical methods for solving zero-sum finite-horizon differential games. We use some results from these papers for additional verification of algorithms' performance in our experiments. Note that these methods are mainly capable of solving low-dimensional differential games and cannot be scaled due to the curse of dimensionality. Thus, the research presented in this paper contributes not only to algorithms effectively solving RRL problems but also to the field of heuristic numerical methods for solving complex zero-sum differential games.

## 2 POSITIONAL DIFFERENTIAL GAMES

Recent studies consider RRL problems within the framework of zero-sum Markov games in pure or mixed policies. In the case of pure policies, it is known (e.g., paper-rock-scissors) that Markov games may not have a value (Nash equilibrium), which conceptually prevents the development of centralized learning algorithms based on shared q-functions. In the case of mixed policies, such formalization may also be inappropriate if, according to the problem statement (for example, in the case of developing expensive or safe control systems), it is required to seek robust policies guaranteeing a deterministic payoff value. In this section, we describe the framework of the positional differential games, which allows us, on the one hand, to consider the pure agents' policies and deterministic values of payoffs and, on the other hand, to obtain the fact of the existence of a value in a reasonably general case.

Let $(\tau, w) \in [0, T) \times \mathbb{R}^n$. Consider a finite-horizon zero-sum differential game described by the differential equation

$$\frac{d}{dt}x(t) = f(t, x(t), u(t), v(t)), \quad t \in [\tau, T], \tag{1}$$

with the initial condition $x(\tau) = w$ and the quality index

$$J = \sigma(x(T)) + \int_\tau^T f^0(t, x(t), u(t), v(t))dt. \tag{2}$$

Here $t$ is a time variable; $x(t) \in \mathbb{R}^n$ is a vector of the motion; $u(t) \in \mathcal{U}$ and $v(t) \in \mathcal{V}$ are control actions of the first and the second agents, respectively; $\mathcal{U} \subset \mathbb{R}^k$ and $\mathcal{V} \subset \mathbb{R}^l$ are compact sets; the function $\sigma(x)$, $x \in \mathbb{R}^n$ is continuous; the functions $f(t, x, u, v)$ and $f^0(t, x, u, v)$ satisfy conditions (see Appendix A for details) sufficient to ensure the existence and uniqueness of a motion $x(\cdot)$ for each pair of Lebesgue measurable functions $(u(\cdot), v(\cdot))$.

The first agent, utilizing the actions $u(t)$, tends to minimize $J$ (see (2)), while the second agent aims to maximize $J$ utilizing the actions $v(t)$.

Following the positional approach to the differential game formalism (Krasovskii & Subbotin (1987); Subbotin (1995)), let us define the following mathematical constructions. Denote

$$\Delta = \big\{t_i \colon t_0 = \tau, \ t_i < t_{i+1}, \ i = 0, 1, \ldots, m, \ t_{m+1} = T\big\},$$
$$d(\Delta) = \max_{i=0,1,\ldots,m} \Delta t_i, \quad \Delta t_i = t_{i+1} - t_i, \quad i = 0, 1, \ldots, m. \tag{3}$$

By a policy of the first agent, we mean an arbitrary function $\pi_u \colon [0, T] \times \mathbb{R}^n \mapsto \mathcal{U}$. Then the pair $\{\pi_u, \Delta\}$ defines a control law that forms the piecewise constant (and therefore, measurable) function $u(\cdot)$ according to the step-by-step rule

$$u(t) = \pi_u(t_i, x(t_i)), \quad t \in [t_i, t_{i+1}), \quad i = 0, 1, \ldots, m.$$

This law, together with a function $v(\cdot)$, uniquely determines the quality index value $J$ (2). The guaranteed result of the policy $\pi_u$ and the optimal guaranteed result of the first agent are defined as

$$V_u^{\pi_u}(\tau, w) = \lim_{\delta \to 0+} \sup_{\Delta: \ d(\Delta) \leq \delta} \sup_{v(\cdot)} J, \quad V_u(\tau, w) = \inf_{\pi_u} V_u^{\pi_u}(\tau, w). \tag{4}$$

Similarly, for the second agent, we consider a policy $\pi_v \colon [0, T] \times \mathbb{R}^n \mapsto \mathcal{V}$, control law $\{\pi_v, \Delta\}$ that forms actions $v(t)$ and define the guaranteed result of $\pi_v$ and the optimal guaranteed result as

$$V_v^{\pi_v}(\tau, w) = \lim_{\delta \to 0+} \inf_{\Delta: \ d(\Delta) \leq \delta} \inf_{u(\cdot)} J, \quad V_v(\tau, w) = \sup_{\pi_v} V_v^{\pi_v}(\tau, w). \tag{5}$$

The fundamental fact (presented as Theorem 12.3 in Subbotin (1995)) of the positional differential game theory on which we rely to obtain our theoretical results (see Theorem 1 below) is as follows: if the functions $f$ and $f^0$ satisfy Isaacs's condition (or the saddle point condition in a small game (Krasovskii & Subbotin (1987)) in other terminology)

$$\min_{u \in \mathcal{U}} \max_{v \in \mathcal{V}} \left( \langle f(t, x, u, v), s \rangle + f^0(t, x, u, v) \right) = \max_{v \in \mathcal{V}} \min_{u \in \mathcal{U}} \left( \langle f(t, x, u, v), s \rangle + f^0(t, x, u, v) \right) \tag{6}$$

for any $t \in [0, T]$ and $x, s \in \mathbb{R}^n$, then, differential game (1), (2) has a value (Nash equilibrium):

$$V(\tau, w) = V_u(\tau, w) = V_v(\tau, w), \quad (\tau, w) \in [0, T] \times \mathbb{R}^n.$$

Note that the important consequence from this fact is that if the equality (6) holds, then any additional knowledge for agents, for example, about the history of the motion $x(\xi)$, $\xi \in [0, t]$, or opponent current actions, does not improve their optimal guaranteed results. Thus, the control laws $\{\pi_u, \Delta\}$ and $\{\pi_v, \Delta\}$ are sufficient to solve zero-sum differential games optimally.

We also note that in order to obtain further results, it is essential not only the existence of a value but also the fulfilment of Isaacs's condition as such.

## 3 SHARED Q-FUNCTION FOR APPROXIMATE BELLMAN EQUATIONS

First of all, to solve differential games by RL algorithms, it is necessary to discretize them in time. In this section, we describe such a discretization, introduce the corresponding game-theoretic constructions, discuss their connection with the Markov game theory, and present the main theoretical result of the paper (Theorem 1).

Let us fix a partition $\Delta$ (3). For each $(t_i, x) \in \Delta \times \mathbb{R}^n$, $i \neq m + 1$, consider the discrete-time differential game (Fleming (1961); Friedman (1971))

$$x_i = x, \quad x_{j+1} = x_j + \Delta t_j f(t_j, x_j, u_j, v_j), \quad u_j \in \mathcal{U}, \quad v_j \in \mathcal{V}, \quad j = i, i+1, \ldots, m,$$

$$J^\Delta = \sigma(x_{m+1}) + \sum_{j=i}^m \Delta t_j f^0(t_j, x_j, u_j, v_j). \tag{7}$$

Note that this game can be formalized (see Appendix B) as a Markov game $(S^\Delta, \mathcal{U}, \mathcal{V}, \mathcal{P}^\Delta, \mathcal{R}^\Delta, \gamma)$, where $S^\Delta$ is a state space consisting of the states $s = (t_i, x) \in \Delta \times \mathbb{R}^n$, $\mathcal{U}$ and $\mathcal{V}$ are the action spaces of the first and second agents, respectively, $\mathcal{P}^\Delta$ is the transition distribution which is a delta distribution in the case under consideration, $\mathcal{R}^\Delta$ is the reward function, $\gamma = 1$ is a discount factor.

Define the pure agents' policies as $\pi_u^\Delta \colon S^\Delta \mapsto \mathcal{U}$ and $\pi_v^\Delta \colon S^\Delta \mapsto \mathcal{V}$, their guaranteed results as

$$V_u^{\pi_u^\Delta}(t_i, x) = \max_{v_i, \ldots, v_m} \left\{ J^\Delta \colon \ x_i = x, \ x_{j+1} = x_j + \Delta t_j f(t_j, x_j, \pi_u^\Delta(t_j, x_j), v_j), \ j = i, \ldots, m \right\},$$

$$V_v^{\pi_v^\Delta}(t_i, x) = \min_{u_i, \ldots, u_m} \left\{ J^\Delta \colon \ x_i = x, \ x_{j+1} = x_j + \Delta t_j f(t_j, x_j, u_j, \pi_v^\Delta(t_j, x_j)), \ j = i, \ldots, m \right\},$$

and the optimal action-value functions (Q-functions) for each agent as

$$Q_u^\Delta(t_i, x, u, v) = r + \inf_{\pi_u^\Delta} V_u^{\pi_u^\Delta}(t_{i+1}, x'), \quad Q_v^\Delta(t_i, x, u, v) = r + \sup_{\pi_v^\Delta} V_v^{\pi_v^\Delta}(t_{i+1}, x'),$$

where $r = \Delta t_i f^0(t_i, x, u, v)$ and $x' = x + \Delta t_i f(t_i, x, u, v)$. Due to this definition, one can show these Q-functions satisfy the following Bellman optimality equations:

$$Q_u^\Delta(t_i, x, u, v) = r + \min_{u' \in \mathcal{U}} \max_{v' \in \mathcal{V}} Q_u^\Delta(t_{i+1}, x', u', v'),$$
$$Q_v^\Delta(t_i, x, u, v) = r + \max_{v' \in \mathcal{V}} \min_{u' \in \mathcal{U}} Q_v^\Delta(t_{i+1}, x', u', v'). \tag{8}$$

It is known that the equality $V_u^\Delta(t_i, z) = V_v^\Delta(t_i, z)$ and as a consequence, the equality $Q_u^\Delta(t_i, x, u, v) = Q_v^\Delta(t_i, x, u, v)$ does not hold in even the simplest Markov game examples (e.g., paper-rock-scissors game). Nevertheless, the fact that this Markov game arises due to a time-discretization of a differential game having a value allowed us to obtain the following crucial result.

**Theorem 1.** Let Isaacs's condition (6) holds. Let the value function $V(\tau, w)$ be continuously differentiable at every $(\tau, w) \in [0, T] \times \mathbb{R}^n$. Then the following statements are valid:

a) For every compact set $D \subset \mathbb{R}^n$ and $\varepsilon > 0$, there exists $\delta > 0$ with the following property. For every partition $\Delta$ satisfying $diam(\Delta) \leq \delta$, there exists a continuous function $Q^\Delta(t_i, x, u, v)$, $(t_i, x, u, v) \in \Delta \times \mathbb{R}^n \times \mathcal{U} \times \mathcal{V}$, such that

$$\left| Q^\Delta(t_i, x, u, v) - r - \min_{u' \in \mathcal{U}} \max_{v' \in \mathcal{V}} Q^\Delta(t_{i+1}, x', u', v') \right| \leq \Delta t_i \varepsilon,$$
$$\left| Q^\Delta(t_i, x, u, v) - r - \max_{v' \in \mathcal{V}} \min_{u' \in \mathcal{U}} Q^\Delta(t_{i+1}, x', u', v') \right| \leq \Delta t_i \varepsilon \tag{9}$$

for any $(t_i, x) \in \Delta \times D$, $i \neq m + 1$, $u \in \mathcal{U}$, $v \in \mathcal{V}$, where $r = \Delta t_i f^0(t_i, x, u, v)$ and $x' = x + \Delta t_i f(t_i, x, u, v)$ and we put $Q^\Delta(t_{m+1}, x', u', v') = \sigma(t_{m+1})$.

Moreover, if the functions $f$ and $f^0$ have the form

$$f(t, x, u, v) = f_u(t, x, u) + f_v(t, x, v), \quad f^0(t, x, u, v) = f_u^0(t, x, u) + f_v^0(t, x, v), \tag{10}$$

then there exists the functions $Q_1^\Delta(t_i, x, u)$ and $Q_2^\Delta(t_i, x, v)$ such that

$$Q^\Delta(t_i, x, u, v) = Q_1^\Delta(t_i, x, u) + Q_2^\Delta(t_i, x, v), \quad (t_i, x, u, v) \in \Delta \times \mathbb{R}^n \times \mathcal{U} \times \mathcal{V}.$$

b) Let $(\tau, w) \in [0, T) \times \mathbb{R}^n$ and $\varepsilon > 0$. There exists a compact set $D \subset \mathbb{R}^n$ and $\delta > 0$ with the following property. For every partition $\Delta$ satisfying $diam(\Delta) < \delta$, there exists a continuous function $Q^\Delta(t_i, x, u, v)$, $(t_i, x, u, v) \in \Delta \times \mathbb{R}^n \times \mathcal{U} \times \mathcal{V}$ satisfying (9), such that the policies

$$\pi_u^\Delta(t_i, x) = \operatorname*{Argmin}_{u \in \mathcal{U}} \max_{v \in \mathcal{V}} Q^\Delta(t_i, x, u, v), \quad \pi_v^\Delta(t_i, x) = \operatorname*{Argmax}_{v \in \mathcal{V}} \min_{u \in \mathcal{U}} Q^\Delta(t_i, x, u, v), \tag{11}$$

provide the inequalities $V_u^{\pi_u^\Delta}(\tau, w) \leq V(\tau, w) + \varepsilon T$ and $V_v^{\pi_v^\Delta}(\tau, w) \geq V(\tau, w) - \varepsilon T$.

**Corollary 1.** Let Isaacs's condition (6) holds. Let the value function $V(\tau, w)$ be continuously differentiable at every $(\tau, w) \in [0, T] \times \mathbb{R}^n$. If the finite sets $\mathcal{U}_*$ and $\mathcal{V}_*$ satisfy the equality

$$\min_{u \in \mathcal{U}_*} \max_{v \in \mathcal{V}_*} \left( \langle f(t, x, u, v), s \rangle + f^0(t, x, u, v) \right) = \min_{u \in \mathcal{U}} \max_{v \in \mathcal{V}} \left( \langle f(t, x, u, v), s \rangle + f^0(t, x, u, v) \right),$$
$$\max_{v \in \mathcal{V}_*} \min_{u \in \mathcal{U}_*} \left( \langle f(t, x, u, v), s \rangle + f^0(t, x, u, v) \right) = \max_{v \in \mathcal{V}} \min_{u \in \mathcal{U}} \left( \langle f(t, x, u, v), s \rangle + f^0(t, x, u, v) \right),$$

then these sets can be used instead of $\mathcal{U}$ and $\mathcal{V}$ in all statements of Theorem 1.

The proof of the theorem and the corollary are given in Appendix C

Theorem 1 shows that there exists a function $Q^\Delta$ which is an approximate solution of both minimax and maximin Bellman optimality equations (8), and this shared Q-function can be exploited to construct policies providing near-optimal guaranteed results in differential games (1), (2). Besides, if the condition (10) holds, then the dependence of $Q^\Delta$ on the agents' actions $u$ and $v$ can be separated. Corollary 1 allows us to approximate the sets $\mathcal{U}$ and $\mathcal{V}$ by finite sets $\mathcal{U}_*$ and $\mathcal{V}_*$ without losing the theoretical results of the theorem, which is essential for further developing algorithms.

## 4 TWO-AGENT DEEP Q-NETWORKS

In this section, we describe various approaches to extending the DQN algorithm (Mnih et al. (2015)) to differential games (1), (2), considering ideas from previous research and the results of Theorem 1.

**NashDQN.** First, we consider the NashDQN (Ding et al. (2022)) (or the similar MinimaxDQN (Fan et al. (2020))) algorithm naturally extending DQN to zero-sum Markov games. Both agents utilize a shared Q-function approximated by a neural network $Q^\theta(t, x, u, v)$. The input of the neural network is $(t, x)$, and the output is the matrix in which the rows correspond to $u$, and the columns correspond to $v$. Agents act according to the $\zeta$-greedy mixed policies

$$u_i \sim (1 - \zeta)\pi_u^\theta(\cdot \mid t_i, x_i) + \zeta\pi_u^{uniform}(\cdot), \quad v_i \sim (1 - \zeta)\pi_v^\theta(\cdot \mid t_i, x_i) + \zeta\pi_v^{uniform}(\cdot),$$

where $\pi_u^\theta(\cdot \mid t_i, x_i)$ and $\pi_v^\theta(\cdot \mid t_i, x_i)$ are the optimal mixed policies in the matrix game $Q^\theta(t_i, x_i, \cdot, \cdot)$, $\zeta \in [0, 1]$ is an exploration parameter, and store $(t_i, x_i, u_i, v_i, r_i, t_{i+1}, x_{i+1})$ into the buffer $\mathcal{D}$. Simultaneously, the minibatch $\{(t_j, x_j, u_j, v_j, r_j, t_j', x_j')\}_{j=1}^{n_{bs}}$ is taken from $\mathcal{D}$ and the loss function

$$L(\theta) = \frac{1}{n_{bs}} \sum_{j=1}^{n_{bs}} \Big( Q^\theta(t_j, x_j, u_j, v_j) - r_j - \gamma V^\theta(t_{j+1}, x_{j+1}) \Big),$$

is utilized to update $\theta$, where $V^\theta(t_{j+1}, x_{j+1})$ is a Nash equilibrium of the matrix game $Q^\theta(t_{j+1}, x_{j+1}, \cdot, \cdot)$. During the learning, the exploration parameter $\zeta$ is decreasing from 1 to 0. Thus, after learning, we obtain mixed policies $\pi_u^\theta(\cdot \mid t, x)$ and $\pi_v^\theta(\cdot \mid t, x)$.

**Multi-agent DQN (MADQN).** Next, we propose to consider another natural extension of the DQN algorithm following the idea from Lowe et al. (2017). Each agent uses its own Q-function approximated by neural networks $Q^{\theta_u}(t, x, u, v)$ and $Q^{\theta_v}(t, x, u, v)$ for the first agent and the second agent, respectively, and act according to the $\zeta$-greedy policies choosing greedy actions

$$\begin{aligned}
u_i = \pi_u^\theta(t_i, x_i) &:= \operatorname*{Argmin}_{u \in \mathcal{U}_*} \max_{v \in \mathcal{V}_*} Q^{\theta_u}(t_i, x_i, u, v), \\
v_i = \pi_v^\theta(t_i, x_i) &:= \operatorname*{Argmax}_{v \in \mathcal{V}_*} \min_{u \in \mathcal{U}_*} Q^{\theta_v}(t_i, x_i, u, v)
\end{aligned} \tag{12}$$

with probability $1 - \zeta$ and any action uniformly on $\mathcal{U}_*$ and $\mathcal{V}_*$ with probability $\zeta$. For learning on the minibatches $\{(t_j, x_j, u_j, v_j, r_j, t_j', x_j')\}_{j=1}^{n_{bs}}$, each of them uses its own loss function

$$\begin{aligned}
L_u(\theta_u) &= \frac{1}{n_{bs}} \sum_{j=1}^{n_{bs}} \Big( Q^{\theta_u}(t_j, x_j, u_j, v_j) - r_j - \gamma \min_{u' \in \mathcal{U}_*} \max_{v' \in \mathcal{V}_*} Q^{\theta_u}(t_j', x_j', u', v') \Big)^2, \\
L_v(\theta_v) &= \frac{1}{n_{bs}} \sum_{j=1}^{n_{bs}} \Big( Q^{\theta_v}(t_j, x_j, u_j, v_j) - r_j - \gamma \max_{v' \in \mathcal{V}_*} \min_{u' \in \mathcal{U}_*} Q^{\theta_v}(t_j', x_j', u', v') \Big)^2.
\end{aligned} \tag{13}$$

In this case, we obtain pure agents' policies $\pi_u^\theta(t, x)$ and $\pi_v^\theta(t, x)$. Note that this algorithm is also centralized since the agents have a shared memory containing opponent actions.

**CounterDQN.** Following the idea from Li et al. (2022), we can complicate the first agent's learning by assuming that the second agent knows its next action in advance. In this case, in contrast to MADQN, the greedy second agent's policy is $\pi_v^\theta(t_i, x_i, u_i) = \operatorname{Argmax}_{v \in \mathcal{V}_*} Q^{\theta_u}(t_i, x_i, u_i, v)$, and hence, only the first Bellman equation in (8) needs to be solved, i.e., only the first loss function in (13) must be minimized. After learning, we obtain the first agent's pure policy $\pi_u^\theta(t, x)$ and the second agent's counter policy $\pi_v^\theta(t, x, u)$. Thus, if we want to obtain the second agent's pure strategy $\pi_v^\theta(t, x)$, we must conduct symmetric learning for them.

**Isaacs's DQN (IDQN).** Now, we modify MADQN utilizing the approximation $Q^\theta(t, x, u, v)$ for the shared Q-function $Q^\Delta(t, x, u, v)$ from Theorem 1. Then, the agents' actions are chosen similar to MADQN, in which $Q^{\theta_u}$ and $Q^{\theta_u}$ are replaced by $Q^\theta$, and the parameter vector $\theta$ is updated according to the loss function

$$L(\theta) = \frac{1}{n_{bs}} \sum_{j=1}^{n_{bs}} \big( Q^\theta(t_j, x_j, u_j, v_j) - y_j \big)^2,$$

where

$$y_j = r_j + \frac{\gamma}{2} \Big( \min_{u \in \mathcal{U}_*} \max_{v \in \mathcal{V}_*} Q^\theta(t_j, x_j, u, v) + \max_{v \in \mathcal{V}_*} \min_{u \in \mathcal{U}_*} Q^\theta(t_j, x_j, u, v) \Big). \tag{14}$$

We use this formula to provide symmetrical learning for the agents since $Q^\theta$ may not satisfy the equality $\min_{u \in \mathcal{U}_*} \max_{v \in \mathcal{V}_*} Q^\theta(t_j, x_j, u, v) = \max_{v \in \mathcal{V}_*} \min_{u \in \mathcal{U}_*} Q^\theta(t_j, x_j, u, v)$.

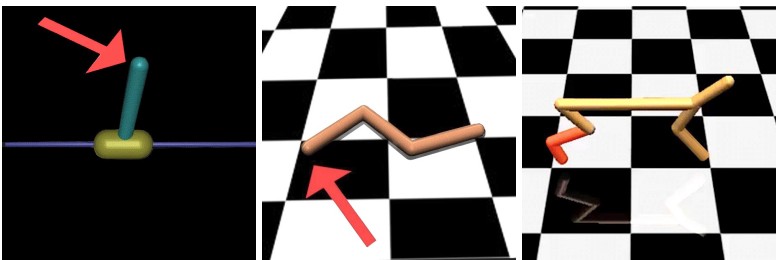

Figure 1: Visualization of the second agent's actions in the games based on the MuJoCo tasks

**Decomposed Isaacs's DQN (DIDQN)**    Finally, according to Theorem 1, we can approximate the function $Q^\Delta(t, x, u, v)$ by the network $Q^\theta(t, x, u, v) = Q^{\theta_1}(t, x, u) + Q^{\theta_2}(t, x, v)$, simplifying calculations of minimums and maximums in (12) and (14) as well as the learning on the whole.

## 5 EXPERIMENTS

**Algorithms.**    In our experiments, we test the following algorithms: the DDQN algorithm (van Hasselt et al. (2016)) for decentralized (simultaneous) learning (2xDDQN) as the most straightforward approach to solve any multi-agent tasks; the PPO algorithm (Schulman et al. (2017)) for alternating learning proposed in Pinto et al. (2017) as an approach for solving RARL problems (RARL); the MADDPG algorithm from Lowe et al. (2017); the NashDQN, MADQN, CounterDQN, IDQN, DIDQN algorithms described above. The algorithms' parameters are detailed in Appendix D.

**Environments.**    We consider the following five zero-sum differential games. *EscapeFromZero* ($\mathcal{S} = \mathbb{R}^2, \mathcal{U} \subset \mathbb{R}^2, \mathcal{V} \subset \mathbb{R}^2$) is presented in Subbotin (1995) as an example of a game where the first agent can move away from zero more than $0.5$ only utilizing a discontinuous policy (V=-0.5). In *GetIntoCircle* ($\mathcal{S} = \mathbb{R}^3, \mathcal{U} \subset \mathbb{R}^1, \mathcal{V} \subset \mathbb{R}^1$) from Kamneva (2019) and *GetIntoSquare* ($\mathcal{S} = \mathbb{R}^3$, $\mathcal{U} \subset \mathbb{R}^1, \mathcal{V} \subset \mathbb{R}^1$) from Patsko (1996), the first agent tends to be as close to zero as possible, but these papers show that the best results it can guarantee to achieve are to be on the border of a circle ($V = 0$) and a square ($V = 1$), respectively. *HomicidalChauffeur* ($\mathcal{S} = \mathbb{R}^5, \mathcal{U} \subset \mathbb{R}^1, \mathcal{V} \subset \mathbb{R}^1$) and *Interception* ($\mathcal{S} = \mathbb{R}^{11}, \mathcal{U} \subset \mathbb{R}^2, \mathcal{V} \subset \mathbb{R}^2$) are the games from Isaacs (1965) and Kumkov et al. (2005), in which the first player wants to be as close as possible to the second agent at the terminal time. We also consider three games based on Mujoco tasks from Todorov et al. (2012), in which we introduce the second agent acting on the tip of the rod in *InvertedPendulum* ($\mathcal{S} = \mathbb{R}^5, \mathcal{U} \subset \mathbb{R}^1$, $\mathcal{V} \subset \mathbb{R}^1$), on the tail of *Swimmer* ($\mathcal{S} = \mathbb{R}^9, \mathcal{U} \subset \mathbb{R}^2, \mathcal{V} \subset \mathbb{R}^1$), or controlling the rear bottom joint of *HalfCheetah* ($\mathcal{S} = \mathbb{R}^{18}, \mathcal{U} \subset \mathbb{R}^5, \mathcal{V} \subset \mathbb{R}^1$) (see Fig. 1). A detailed description of all the above games is provided in Appendix E.

**Evaluation scheme.**    We consider the following evaluation scheme in our experiments (see Fig. 2). In the first stage, agents learn (decentralized or centralized, depending on an algorithm). In the second stage, we fix the trained first agent's policy $\pi_u$ and solve the obtained single-agent RL problem from the point of the second agent's view using various baseline RL algorithms with various hyperparameters (see Appendix D for details). After that, we choose the maximum value of quality index $J$ (2) (sum of rewards) in these running and put it into the array "maximum values of the quality index". We believe this maximum value approximates the guaranteed result $V_u^{\pi_u}$ (4). The third step is symmetrical to the second one and is aimed at obtaining an approximation for $V_v^{\pi_v}$ (5). We repeat these three stages 5 times, accumulating "maximum values of the quality index" and "minimum values of the quality index" arrays. Then, we illustrate the data of these arrays as shown in Fig. 2. The boldest bar describes the best guaranteed results of the agents out of 5 runnings, the middle bar gives us the mean values, and the faintest bar shows the worst results in 5 runnings. The width of the bars illustrates the exploitability of both agents, that is, the difference between the obtained approximations of $V_u^{\pi_u}$ and $V_v^{\pi_v}$. If they are close to the optimal guaranteed results $V_u$ and $V_v$, then the width should be close to zero (if a value exists ($V_u = V_v$)). Thus, looking at such a visualization, we can make conclusions about the stability (with respect to running) and efficiency (with respect to exploitability) of the algorithms.

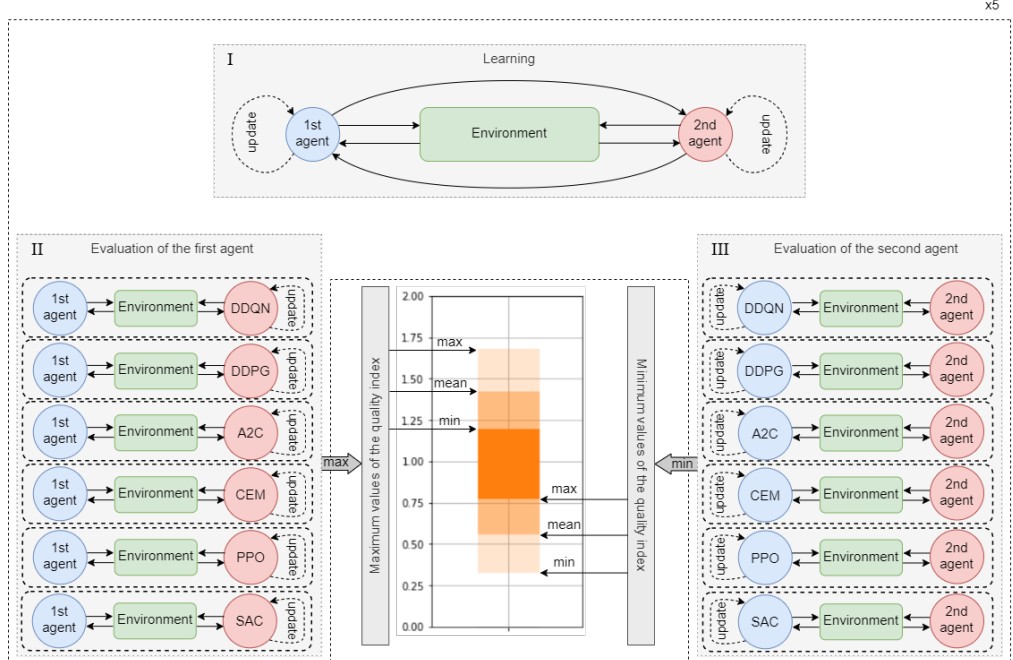

Figure 2: Evaluation scheme.

**Experimental results.** Fig. 3 shows the experimental results of the algorithms and the accurate values (dotted line) when we know them. First of all, we note that the 2xDDQN, RARL, and NashDQN algorithms show the worst performance. In the case of 2xDDQN and RARL, the reason is quite typical for decentralized learning. An agent overfits against a specific opponent and loses the ability to resist other opponents. In the case of NashDQN, the reason, apparently, lies in the stochasticity of the trained policies aimed at giving results on average but not guaranteed.

The MADDPG algorithm demonstrates the satisfactory borders of guaranteed results only in 2 games (GetInfoCircle and GetInfoSquare). Regarding average by runnings, the algorithm is also well in HomicidalChauffeur, InvertedPendulum, and Swimmer, which reflects, on the one hand, the potential ability of MADDPG to find policies close to optimal, but, on the other hand, its instability with respect to running.

The MADQN algorithm is generally better than the algorithms discussed above, but it still inferiors to IDQN and DIDQN in all games.

The CounterDQN algorithm gives worse results than MADQN in almost all games (except HomicidalChauffeur and InvertedPendulum), which apparently indicates that it is more efficient for agents to have more learning time steps than information about the opponent's actions.

The IDQN and DIDQN algorithms show the best performance in all games, reflecting the advantage of utilizing a shared Q-function. These algorithms show similar performance except InvertedPendulum where DIDQN is clearly better.

Thus, we conclude the following: centralized learning is much more efficient than decentralized learning, solving the Bellman equation in pure policies gives better results than in mixed ones, a shared Q-function makes learning more stable than two independent Q-functions, and the Q-function decomposition can provide an advantage in some tasks.

## 6 LIMITATIONS

Although Isaacs's condition is quite common and can often be verified by relying only on general ideas about dynamics, there are cases when it is not fulfilled. In these cases, Theorem 1 is not valid, and therefore, it seems more theoretically justified to use MADQN instead of IDQN and DIDQN.

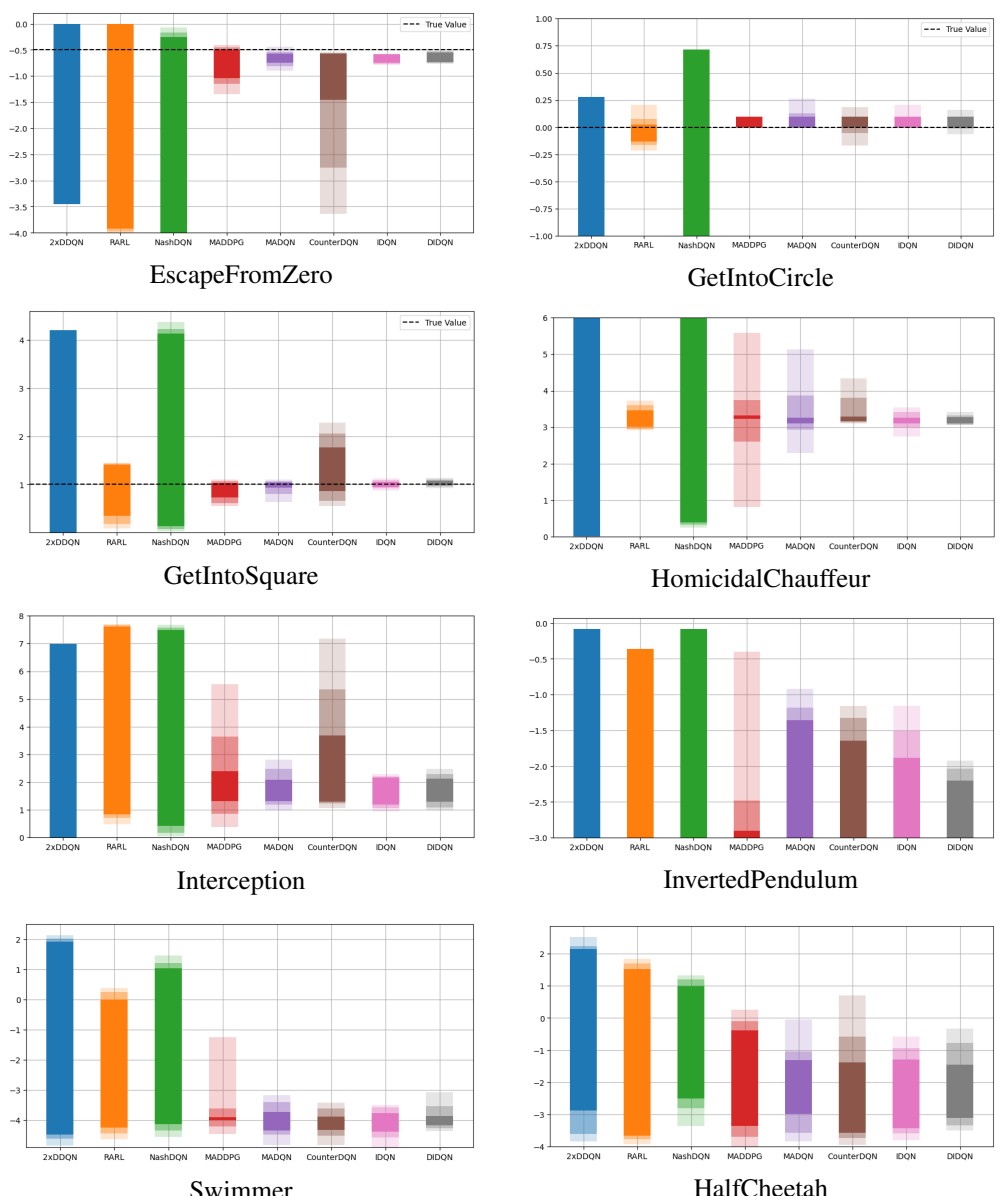

Figure 3: Experimental results.

An essential limitation of MADQN, IDQN, and DIDQN, as well as the basic DQN, is the action space's finiteness. In our paper, we show (see Corollary 1) that the action space discretization leaves the results of Theorem 1 valid under certain conditions. However, modifying the proposed algorithms for continuous action space is a promising direction for further research that can improve their performance, especially for high-dimensional action spaces.

The proposed IDQN and DIDQN algorithms can be interpreted not only as algorithms for solving RRL problems but also as algorithms for solving zero-sum differential games. In this sense, it should be emphasized that the development of the shared Q-function concept to the general case of multi-agent differential games is non-trivial and is complicated by the fact that there are no simple and sufficiently general conditions (analogous to Isaacs's condition) under which such games have an equilibrium in positional (feedback) policies. Nevertheless, in some classes of games in which the existence of Nash equilibrium is established, such investigations can be promising.

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

## A    APPENDIX

Typical conditions for the positional differential game theory (see, e.g., p. 116 in Subbotin (1995)) are the following:

- The functions $f(t, x, u, v)$ and $f^0(t, x, u, v)$ are continuous.
- There exists $c_f > 0$ such that

$$\big\| f(t, x, u, v) \big\| + \big| f^0(t, x, u, v) \big| \leq c_f \big( 1 + \|x\| \big), \quad (t, x, u, v) \in [0, T] \times \mathbb{R}^n \times \mathcal{U} \times \mathcal{V}.$$

- For every $\alpha > 0$, there exists $\lambda_f > 0$ such that

$$\big\| f(t, x, u, v) - f(t, y, u, v) \big\| + \big| f^0(t, x, u, v) - f^0(t, y, u, v) \big| \leq \lambda_f \|x - y\|$$

for any $t \in [0, T]$, $x, y \in \mathbb{R}^n$: $\max\{\|x\|, \|y\|\} \leq \alpha$, $u \in \mathcal{U}$, and $v \in \mathcal{V}$.

In particular, these conditions provide the existence and uniqueness of the motion $x(\cdot)$ for each Lebesgue-measurable functions $u(\cdot)\colon [\tau, T] \mapsto \mathcal{U}$ and $v(\cdot)\colon [\tau, T] \mapsto \mathcal{V}$, where we mean by the motion a Lipschitz continuity function $x(\cdot)\colon [\tau, T] \mapsto \mathbb{R}^n$ satisfying condition $x(\tau) = w$ and equation (1) almost everywhere.

## B    APPENDIX

Let us show that game (7) can be formalized as a Markov game $(S^\Delta, \mathcal{U}, \mathcal{V}, \mathcal{P}^\Delta, \mathcal{R}^\Delta, \gamma)$. First, put

$$\mathcal{S}^\Delta = \big( \Delta \times \mathbb{R}^n \big) \cup s_T,$$

where $s_T$ is some fictional terminal state. Next, for every $s = (t_i, x) \in \Delta \times \mathbb{R}^n$, $i \neq m + 1$, $u \in \mathcal{U}$, and $v \in \mathcal{V}$, we define the transition distribution and the reward function by

$$\mathcal{P}(s'|s, u, v) = \delta(s'), \quad \mathcal{R}(s, u, v) = \Delta t_i f^0(t_i, x, u, v),$$

where $s' = (t_{i+1}, x')$, $x' = x + \Delta t_i f(t_i, x, u, v)$, and $\delta$ is the Dirac delta distribution. For $s = (t_{m+1}, x) \in \Delta \times \mathbb{R}^n$, we set

$$\mathcal{P}(s'|s, u, v) = \delta(s' = s_T), \quad \mathcal{R}(s, u, v) = \sigma(x), \quad u \in \mathcal{U}, \quad v \in \mathcal{V}.$$

In order to make the game formally infinite, we put

$$\mathcal{P}(s'|s_T, u, v) = \delta(s' = s_T), \quad \mathcal{R}(s_T, u, v) = 0, \quad u \in \mathcal{U}, \quad v \in \mathcal{V}.$$

## C    APPENDIX

Denote

$$\chi(t, x, u, v, s) = \langle f(t, x, u, v), s \rangle + f^0(t, x, u, v).$$

**Lemma 1.** Let condition (6) hold. Let the value function $V(\tau, w) = V_u(\tau, w) = V_v(\tau, w)$ be continuously differentiable at every $(\tau, w) \in [0, T] \times \mathbb{R}^n$. Then the equations

$$\frac{\partial}{\partial \tau} V(\tau, w) + H(\tau, w, \nabla_w V(\tau, w)) = 0, \quad V(T, w) = \sigma(w), \tag{15}$$

hold for any $\tau \in [0, T)$ and $w \in \mathbb{R}^n$, where we denote

$$H(t, x, s) = \min_{u \in \mathcal{U}} \max_{v \in \mathcal{V}} \chi(t, x, u, v, s) = \max_{v \in \mathcal{V}} \min_{u \in \mathcal{U}} \chi(t, x, u, v, s).$$

The lemma follows from two facts: the value function is a minimax (generalized) solution of Cauchy problem (15) (see Theorem 11.4 in Subbotin (1995)) and a continuously differentiable minimax solution is a classical solution (see Section 2.4 in Subbotin (1995)).

Let us prove $a$). Let $\varepsilon > 0$ and $D \subset \mathbb{R}^n$. Let us define a compact set $D'$ so that

$$\big\{ x' = x + (t' - t) f(t, x, u, v)\colon t, t' \in [0, T], \ x \in D, \ u \in \mathcal{U}, \ v \in \mathcal{V} \big\} \subset D'.$$

Since the value function $V$ is continuously differentiable, there exists $\delta > 0$ such that

$$\left| V(t', x') - V(t, x) - (t' - t)(\partial/\partial t)V(t, x) - \langle x' - x, \nabla_x V(t, x) \rangle \right| \le \varepsilon(t' - t)$$

for any $t, t' \in [0, T]$ satisfying $0 < t - t' \le \delta$ and any $x \in D$, $x' \in D'$ $u \in \mathcal{U}$, $v \in \mathcal{V}$. Let $\Delta$ be such that $diam(\Delta) < \delta$. Define

$$Q^{\Delta}(t_i, x, u, v) = V(t_i, x) + \Delta t_i \Big( (\partial/\partial t)V(t, x) + \chi(t, x, u, v, \nabla_x V(t, x)) \Big), \qquad (16)$$

where $(t_i, x, u, v) \in \Delta \times \mathbb{R}^n \times \mathcal{U} \times \mathcal{V}$, $i \ne m + 1$, and $Q^{\Delta}(t_{m+1}, x, u, v) = \sigma(x)$. Then, using Lemma 1, we derive

$$\left| Q^{\Delta}(t_i, x, u, v) - r - \min_{u' \in \mathcal{U}} \max_{v' \in \mathcal{V}} Q^{\Delta}(t_{i+1}, x', u', v') \right|$$

$$= \left| V(t_i, x) + \Delta t_i (\partial/\partial t)V(t_i, x) + \langle x' - x, \nabla_x V(t_i, x) \rangle - V(t_{i+1}, x') \right| \le \varepsilon \Delta t_i$$

for any $(t_i, x, u, v) \in \Delta \times \mathbb{R}^n \times \mathcal{U} \times \mathcal{V}$, where $r = \Delta t_i f^0(t_i, x, u, v)$ and $x' = x + \Delta t_i f(t_i, x, u, v)$. The statement about decomposition of $Q^{\Delta}(t, x, u, v)$ follows from (16). Thus, a) has proved.

Let us prove b). Let $(\tau, w) \in [0, T) \times \mathbb{R}^n$ and $\varepsilon > 0$. Put

$$D = \left\{ (t, x) \in [0, T] \times \mathbb{R}^n \colon \|x\| \le (\|w\| + 1)e^{c_f t} - 1 \right\}.$$

Then, we have the inclusion $(\tau, w) \in D$. Note also that, for every $(t, x) \in D$, the inclusion $(t', x') \in D$ holds for $t' \in [t, T]$ and $x' = x + (t' - t)f(t, x, u, v)$, $u \in \mathcal{U}$, $v \in \mathcal{V}$. Take $\delta > 0$ according to a). Let us take a partition $\Delta$ satisfying $diam(\Delta) < \delta$ and the function $Q^{\Delta}(t_i, x, u, v)$ from (16). Let $v_i \in \mathcal{V}$, $i = 0, 1, \ldots, m$ be such that the equality

$$V_u^{\pi_u}(\tau, w) = \sigma(x_{m+1}) + \sum_{i=0}^{m} \Delta t_i f^0(t_i, x_i, \pi_u^{\Delta}(t_i, x_i), v_i),$$

holds, where

$$x_i = w \quad x_{i+1} = x_i + \Delta t_i f(t_i, x_i, \pi_u^{\Delta}(t_i, x_i), v_i), \quad i = 0, 1, \ldots, m.$$

Then, due to (9) and (11), we derive

$$V_u^{\pi_u}(\tau, w) \le \sigma(x_{m+1}) + \sum_{i=0}^{m} \left( Q^{\Delta}(t_i, x_i, \pi_u(t_i, x_i), v_i) - \min_{u \in \mathcal{U}} \max_{v \in \mathcal{V}} Q^{\Delta}(t_{i+1}, x_{i+1}, u, v) \right) + \varepsilon T$$

$$\le \sigma(x_{m+1}) + \sum_{i=0}^{m} \left( \min_{u \in \mathcal{U}} \max_{v \in \mathcal{V}} Q^{\Delta}(t_i, x_i, u, v) - \min_{u \in \mathcal{U}} \max_{v \in \mathcal{V}} Q^{\Delta}(t_{i+1}, x_{i+1}, u, v) \right) + \varepsilon T$$

$$\le \min_{u \in \mathcal{U}} \max_{v \in \mathcal{V}} Q^{\Delta}(\tau, w, u, v) + \varepsilon T$$

Form this estimate, taking into account the definition (16) of $Q^{\Delta}$ and Lemma 1, we obtain the first inequality in the statement b). The second inequality can be proved by the symmetric way.

The validity of Corollary 1 follows from the proof given above replacing

$$\min_{u \in \mathcal{U}} \max_{v \in \mathcal{V}} \chi(t, x, u, v, s), \quad \max_{v \in \mathcal{V}} \min_{u \in \mathcal{U}} \chi(t, x, u, v, s)$$

with

$$\min_{u \in \mathcal{U}_*} \max_{v \in \mathcal{V}_*} \chi(t, x, u, v, s), \quad \max_{v \in \mathcal{V}_*} \min_{u \in \mathcal{U}_*} \chi(t, x, u, v, s).$$

# D  APPENDIX

**Two-agent algorithms' parameters.** We use pretty standard and the same parameters for the 2xDDQN, NasgDQN, MADQN, CounterDQN, IDQN, and DIDQN algorithms. We utilize the ADAM optimazer with the learning rate $lr = 0.001$, the smoothing parameter $\tau = 0.01$, and the batch size $n_{bs} = 64$. For the time-discretization, we use the uniform partitions $\Delta = \{i\Delta t \colon i = $

| Environments | $\mathcal{U}_*$ | $\mathcal{V}_*$ | $\Delta t$ | Hidden NN Leyers |
|---|---|---|---|---|
| EscapeFromZero | $BM(0, 2\pi, 10)$ | $BM(0, 2\pi, 10)$ | 0.2 | $256, 128$ |
| GetIntoCircle | $LM(-0.5, 0.5, 10)$ | $LM(-1, 1, 10)$ | 0.2 | $256, 128$ |
| GetIntoSquare | $LM(-1, 1, 10)$ | $LM(-1, 1, 10)$ | 0.2 | $256, 128$ |
| HomicidalChauffeur | $LM(-1, 1, 10)$ | $BM(0, 2\pi, 10)$ | 0.2 | $256, 128$ |
| Interception | $BM(0, 2\pi, 10)$ | $BM(0, 2\pi, 10)$ | 0.2 | $512, 256, 128$ |
| InvertedPendulum | $LM(-1, 1, 9)$ | $LM(-0.2, 0.2, 9)$ | 0.2 | $512, 256, 128$ |
| Swimmer | $SM(-1, 1, 4, 2)$ | $LM(-0.2, 0.2, 16)$ | 1.0 | $512, 256, 128$ |
| HalfCheetah | $SM(-0.5, 0.5, 2, 5)$ | $LM(-0.5, 0.5, 32)$ | 0.3 | $512, 256, 128$ |

Table 1: Parameters for the two-agents' learning algorithms

| Parameters | DDQN | DDPG | CEM | A2C | PPO | SAC |
|---|---|---|---|---|---|---|
| learning timesteps | 5e4 | 2.5e4 | 5e4 | 2.5e4 | 5e4 | 2.5e4 |
| learning rate | 1e-3 | $\pi$ : 1e-4, $q$ : 1e-3 | 1e-2 | 1e-3 | 1r-3 | 1e-3 |
| batch size | 64 | 64 | — | Def. | 64 | Def. |
| smooth param. $\tau$ | 1e-2 | 1e-3 | 1e-2 | Def. | Def. | 1e-2 |
| discount factor $\gamma$ | 1 | 1 | 1 | 1 | 1 | 1 |
| percentile param. | — | — | 80 | — | — | — |
| number of steps | — | — | — | — | 64 | — |

Table 2: Parameters for the evaluating algorithms.

$0, 1, \ldots, m + 1\}$. The parameter $\Delta t$, the structure of the neural networks, and the discretization of the continuous action spaces depend on the game and are indicated in Table 1, where we define the linear mesh, square mesh, and ball mesh as

$$LM(a, b, k) = \{a + i(b - a)/k \colon i = 0, 1, \ldots, k\}, \quad SM(a, b, k, n) = LM(a, b, k)^n,$$

$$BM(a, b, k) = \{(\sin(\alpha), \cos(\alpha)) \in \mathbb{R}^2 \colon \alpha \in LM(a, b, k)\}.$$

In particular, we use deeper neural networks in more complex games for better results. Agents learn during 50000 timesteps, under the linear reduction of the exploration noise $\zeta$ from 1 to 0. In the CounterDQN algorithm, each agent learns 25000 timesteps.

For the RARL approach, we apply the PPO algorithm from StableBaseline3 with the standard parameters and alternately teach the agents every 1000 timesteps.

**Algorithms' parameters for evaluation.** Parameters of the algorithms used in the evaluation stages (see Fig. 2) are described in Table 2.

# E  APPENDIX

**EscapeFromZero.** The game taken from (Subbotin, 1995, p. 164) describes the motion of a point on a plane that is affected by two agents. The first agent aims to be as far away from zero as possible at the terminal time $T = 2$, while the aim of the second agent is the opposite. The capabilities of the first agent's influence are constant and are described by a unit ball. In contrast, the capabilities of the second agent are a ball with a decreasing radius as the terminal time is approached. Thus, the differential game is described by the differential equation

$$\frac{d}{dt} x(t) = u(t) + (2 - t)v(t), \quad t \in [0, 2], \quad x(t) \in \mathbb{R}^2, \quad u(t), v(t) \in B^2 := \{s \in \mathbb{R}^2 \colon \|s\| \le 1\},$$

with the initial condition $x(0) = x_0 := (0, 0)$, and the quality index $J = -\|x(2)\|$. (Subbotin, 1995, p. 164) shows that $V(0, x_0) = -0.5$. This means the first agent is able to move away from zero by 0.5 at the terminal time $T = 2$ for any actions of the second agent.

**GetIntoCircle**    This game is taken from Kamneva (2019). The first and the second agents can move a point on the plane vertically and horizontally, respectively. The first agent aims to drive the point as close to zero as possible at the terminal time $T = 4$. The aim of the second agent is the opposite. Thus, the differential game is described as follows:

$$\frac{d}{dt}x_1(t) = v(t), \quad \frac{d}{dt}x_2(t) = u(t), \quad t \in [0, 4], \quad x(t) \in \mathbb{R}^2, \quad u(t) \in [-0.5, 0.5], \quad v(t) \in [-1, 1],$$

$$x(0) = x_0 = (0, 0.5), \quad J = \|x(4)\| - 4.$$

This game has a value $V(0, x_0) = 0$, which means the optimal first agent can lead the point only to the border of a circle of the radius $r = 4$.

**GetIntoSquare.**    In the game from Patsko (1996), The first agent aims to drive a point on the plane as close to zero as possible at the terminal time $T = 4$. The aim of the second agent is the opposite. The differential game is described as follows:

$$\frac{d}{dt}x_1(t) = x_2(t) + v(t), \quad \frac{d}{dt}x_2(t) = -x_1(t) + u(t),$$

$$t \in [0, 4], \quad x(t) \in \mathbb{R}^2, \quad u(t), v(t) \in [-1, 1],$$

$$x(0) = x_0 := (0.2, 0), \quad J = \max\{|x_1(4)|, |x_2(4)|\}.$$

The game has the value $V(0, x_0) = 1$, which means the optimal first agents can lead the point only to the border of a square with the side $a = 1$.

**HomicidalChauffeur**    is a well-studied example of a pursuit-evasion differential game (Isaacs (1965)). However, to formalize this game within our class of differential games (1), 2) we consider its finite-horizon version:

$$\frac{d}{dt}x_1(t) = 3\cos(x_3(t)), \quad \frac{d}{dt}x_2(t) = 3\sin(x_3(t)), \quad \frac{d}{dt}x_3(t) = u(t), \quad \frac{d}{dt}x_4(t) = v_1(t),$$

$$\frac{d}{dt}x_5(t) = v_2, \quad t \in [0, 3], \quad x(t) \in \mathbb{R}^5, \quad u(t) \in [-1, 1], \quad v(t) \in \{v \in \mathbb{R}^2 : \|v\| \leq 1\}$$

$$x(0) = (0, 0, 0, 2.5, 7.5), \quad J = \sqrt{(x_1(4) - x_4(t))^2 + (x_2(4) - x_5(4))^2}.$$

Such a version of this game has been studied much less, and therefore, we do not know the exact value in it.

**Interception.**    This game is taken from Kumkov et al. (2005) and describes an air interception task. At the terminal time $T = 3$, the first agent strives to be as close as possible to the second agent, but unlike the second agent, the first agent has inertia in dynamics. The differential game is described by the differential equation

$$\frac{d^2}{dt^2}y(t) = F(t), \quad \frac{d}{dt}F(t) = -F(t) + u(t), \quad \frac{d^2}{dt^2}z(t) = v(t), \quad t \in [0, 3],$$

$$x(t) = \left(y_1(t), y_2(t), \frac{d}{dt}y_1(t), \frac{d}{dt}y_2(t), F_1(t), F_2(t), z_1(t), z_2(t), \frac{d}{dt}z_1(t), \frac{d}{dt}z_2(t)\right) \in \mathbb{R}^{10},$$

$$u(t) \in \left\{u \in \mathbb{R}^2 : \frac{u_1^2}{(0.67)^2} + u_2^2 \leq (1.3)^2\right\}, \quad v(t) \in \left\{v \in \mathbb{R}^2 : \frac{v_1^2}{(0.71)^2} + v_2^2 \leq 1\right\},$$

with the initial conditions $x(0) = x_0 := (1, 1.1, 0, 1, 1, -2, 0, 0, 1, 0)$ and the quality index $J = \|y(3) - z(3)\|$. Due to the difference of the problem statement in Kumkov et al. (2005), we cannot precisely set the value of this game. We can only state the inequality $V(0, x_0) \geq 1.5$.

**InvertedPendulum.**    We take the InvertedPendulum task from the MuJoCo simulator (Todorov et al. (2012)) described by

$$\frac{d}{dt}x(t) = F_{IP}(x(t), u(t)), \quad t \in [0, T_f), \quad x(t) = (qpos_{0:1}(t), qvel_{0:1}(t)) \in \mathbb{R}^4, \quad u(t) \in [-1, 1],$$

where $qpos_{0:1}(t) = (qpos_0(t), qpos_1(t))$, $qvel_{0:1}(t) = (qvel_0(t), qvel_1(t))$, and $T_f$ is a time until which the restriction $x(t) \in D$ holds. Violation of this restriction means the pendulum falls down. Based on this task, we consider the differential game

$$\frac{d}{dt}x(t) = F_{IP}(x(t), u(t)) + e_4 v(t), \ \ t \in [0,3], \ \ x(t) \in \mathbb{R}^4, \ \ u(t) \in [-1,1], \ \ v(t) \in [-0.2, 0.2]$$

where $e_4 = (0,0,0,1)$, with initial condition $x(0) = x_0 = (0,0,0,0)$ and the quality index

$$J = -\int_0^3 [x(t) \in D]dt, \quad [x(t) \in D] = \begin{cases} 1, & \text{if } x(t) \in D \text{ holds}, \\ 0, & \text{otherwise}. \end{cases}$$

Thus, we introduced the second agent as a disturbance at the end of the rod and reformulated the problem as differential game (1), (2), retaining the meaning.

**Swimmer.** In a similar way, we consider the Swimmer task from MuJoCo

$$\frac{d}{dt}x(t) = F_S(x(t), u(t)), \ \ t \in [0, +\infty), \ \ x(t) = (qpos_{2:4}(t), qvel_{0:4}(t)) \in \mathbb{R}^8, \ \ u(t) \in [-1,1]^2,$$

$$x(0) = x_0 = 0 \in \mathbb{R}^8, \quad J = \int_0^{+\infty} r(x(t))dt,$$

introduces the second agent as a disturbance on the tail, and reformulated this task as differential game (1), (2)

$$\frac{d}{dt}x(t) = F_S(x(t), u(t)) + e_3 v(t), \ \ t \in [0, 20], \ \ x(t) \in \mathbb{R}^8, \ \ u(t) \in [-1,1]^2, \ \ v(t) \in [-0.2, 0.2],$$

$$x(0) = x_0 = 0 \in \mathbb{R}^8, \quad J = -\int_0^{20} r(x(t))dt.$$

**HalfCheetah** is the third task from the MuJoCo simulator that we consider

$$\frac{d}{dt}x(t) = F_{HC}(x(t), a(t)), \ t \in [0, +\infty), \ x(t) = (qpos_{1:8}(t), qvel_{0:1}(t)) \in \mathbb{R}^{17}, \ a(t) \in [-1,1]^6,$$

$$x(0) = x_0 = 0 \in \mathbb{R}^{17}, \quad J = \int_0^{+\infty} r(x(t))dt.$$

In this task, we determine the agents' actions as $u(t) = a_{1:5}(t) \in [-0.5, 0.5]^5$ and $v(t) = a_6(t) \in [-0.5, 0.5]$ and reformulated the task as differential game (1), (2)

$$\frac{d}{dt}x(t) = F_{HC}(x(t), u(t), v(t)), \ t \in [0,3), \ x(t) \in \mathbb{R}^{17}, \ u(t) \in [-0.5, 0.5]^5, \ v(t) \in [-0.5, 0.5],$$

$$x(0) = x_0 = 0 \in \mathbb{R}^{17}, \quad J = -\int_0^3 r(x(t))dt.$$

Here we reduce the capabilities of agents in order to make the game more interesting. Otherwise, the second agent would always win by flipping the HalfCheetah.

