# OpenReview forum: "Zero-Sum Positional Differential Games as a Framework for Robust Reinforcement Learning: Deep Q-Learning Approach"
_ICLR.cc/2024/Conference — Submitted to ICLR 2024_

### Official Review · Reviewer_Uwp8 · 2023-10-26

**Soundness:** 3 good
**Presentation:** 3 good
**Contribution:** 2 fair
**Rating:** 6
**Confidence:** 3

**Summary:**

This paper studies Robust Reinforcement Learning (RRL) within the framework of positional differential game theory, presenting a novel approach to understanding agents' robust policies and deterministic payoff values. This paper introduces two algorithms: Isaacs Deep Q-Networks (IDQN) and Decomposed Isaacs Deep Q-Networks (DIDQN). Through theoretical proofs, under the Isaacs’s condition, it shows that a centralized Q-learning approach can be developed for these problems. Empirical results underscore the effectiveness of these algorithms against other RRL and Multi-Agent RL baselines, also proposing a new framework for evaluating the robustness of trained policies.

**Strengths:**

Firstly, it offers an exhaustive literature review, meticulously drawing contrasts between the current work and existing literature. Moreover, the clarity of writing facilitates comprehension, making the paper accessible to a broad spectrum of readers. Lastly, the experimental results provided is convincing, adding weight to the authors' arguments and hypotheses.

**Weaknesses:**

1. While the authors adeptly show the significance of considering robustness in RL in the introduction, they do not clearly state why there's a pressing need to study robustness in the framework of positional differential game theory. This oversight makes the paper's motivation less pronounced and leaves readers questioning the specific choice of this framework.

2. One point of confusion for me is in Section 3 where the authors state that "to solve differential games by RL algorithms, it is necessary to discretize them in time，we describe such a discretization...". If employing RL to address differential games still necessitates discretization, then why consider robust RL problems within the framework of differential games? Wouldn't it be more straightforward to tackle the issue directly within the framework of discrete robust MDPs? What additional advantages does considering robustness in differential games provide? Furthermore, how does the efficacy of using discretization to address differential games compare with directly solving a discrete robust MDP?

**Questions:**

Please see the Weaknesses, I will finalize my rating after rebuttal.

**Details Of Ethics Concerns:**

Non

---

> ### Author Response · Authors · 2023-11-16
> **Response**
>
> Dear Reviewer,
>
> Thank you very much for highlighting the positive points of the paper. We are delighted that you liked the literature review, presentation and results. Below, we provide comments regarding your doubts.
>
> ***While the authors adeptly show the significance of considering robustness in RL in the introduction, they do not clearly state why there's a pressing need to study robustness in the framework of positional differential game theory. This oversight makes the paper's motivation less pronounced and leaves readers questioning the specific choice of this framework.***
>
> Thank you for this important comment. We tried to explain the advantage of the differential game concept compared with Markov games in mixed policies in the third paragraph of the Introduction section and compared with Markov games in pure policies before Theorem 1. Perhaps we did this not fully articulated, and to correct this, we added one paragraph to the beginning of section 2, further explaining the motivation to use differential games. Its contents are as follows:
>
> *Recent studies consider RRL problems within the framework of zero-sum Markov games in pure or mixed policies. In the case of pure policies, it is known (e.g., paper-rock-scissors) that Markov games may not have a value (Nash equilibrium), which conceptually prevents the development of centralized learning algorithms based on shared q-functions. In the case of mixed policies, such formalization may also be inappropriate if, according to the problem statement (for example, in the case of developing expensive or safe control systems), it is required to seek robust policies guaranteeing a deterministic payoff value. In this section, we describe the framework of the positional differential games, which allows us, on the one hand, to consider the pure agents' policies and deterministic values of payoffs and, on the other hand, to obtain the fact of the existence of a value in a reasonably general case.*
>
> ***One point of confusion for me is in Section 3 where the authors state that "to solve differential games by RL algorithms, it is necessary to discretize them in time，we describe such a discretization...". If employing RL to address differential games still necessitates discretization, then why consider robust RL problems within the framework of differential games? Wouldn't it be more straightforward to tackle the issue directly within the framework of discrete robust MDPs? What additional advantages does considering robustness in differential games provide?***
>
> We are not sure that there is a well-established concept of robust MDP. Basically, Markov games with pure or mixed policies are taken as such. We tried to describe the advantage of differential games compared with both of these options in the answer to the previous question. If you mean Markov games in pure policies described in Section 3 by robust MDP, then, as we noted above and in the paper (before Theorem 1), their main drawback is the non-existence of value (Nash equilibrium) in general case, which conceptually prevents the use of a shared q-function. However, as we show in Theorem 1, the fact that we consider not arbitrary games but the games which are a time-discretization of differential games satisfying Isaacs's condition allows us to theoretically justify that with a sufficiently small time-discretization step, the shared q-function can be used. If you meant something else by robust MDP, please feel free to clarify, and we will also be glad to discuss this.
>
> ***Furthermore, how does the efficacy of using discretization to address differential games compare with directly solving a discrete robust MDP?***
>
> If we understand your question correctly, then the answer is as follows. In fact, in our experiments, almost all the algorithms we consider aim to solve discrete-time RRL or MARL problems. 2xDDQN represents a naive decentralized learning approach often used in MARL in discrete time. RARL represents the decentralized approach that was used by Pinto et al. (2017) to solve RRL in discrete time. MADDPG is a basic centralized algorithm for solving general MARLs in discrete time. MADQN is its development for the case of a discrete action space. NashDQN is an algorithm aimed at solving Markov games in discrete time. Only CounterDQN and our IDQN and DIDQN are algorithms for which it is theoretically essential that an environment is a discretization of some differential game (in continuous time).
>
> Thank you again for your feedback. We hope that our response has helped to clear up some doubts. If this is the case in your opinion, then we respectfully ask that you consider increasing your score. If you have any more comments, questions or remarks, we would be glad to discuss them.

---

> > ### Comment · Reviewer_Uwp8 · 2023-11-20
> > **Thanks for response**
> >
> > Thanks for the clarification, I have raised my score. Please include all the discussions in the final version.

---

> > > ### Author Response · Authors · 2023-11-21
> > > **Response**
> > >
> > > Thank you very much for raising the score. Of course, we have already updated the paper with all the changes mentioned.

---

### Official Review · Reviewer_BWN5 · 2023-10-31

**Soundness:** 2 fair
**Presentation:** 2 fair
**Contribution:** 2 fair
**Rating:** 3
**Confidence:** 5

**Summary:**

This paper explores robust adversarial reinforcement learning (RARL) through the lens of positional differential game theory, ensuring the worst-case deterministic payoff. Leveraging the positional differential game theory, the authors formulated multi-agent reinforcement learning (MARL) to solve the RARL problem. These techniques were then benchmarked against multiple MARL baseline methods. Finally, the authors analyze the learned policies, highlighting the superior performance of the introduced algorithms.

**Strengths:**

This paper provides a novel perspective of RARL in the context of positional differential games.

**Weaknesses:**

* There is substantial room for improvement in terms of writing. There is a lack of overall organization of sections and smooth transitions between paragraphs. Some examples are:
    * In section 2, the authors could provide more insights into what is the difference between the standard formulation and differential game and why it is important to have a deterministic payoff. How the fact of the PDG is important to this paper and when the Isaacs's condition is met or not met.
    * In the experiment section, the metric of 'stability' appears without proper definition and explanation. For example, is 'stability' equivalent to 'deterministic payoff'? If so, why not just stick to the latter?
    * In the experiment section again, some analysis seems to be out of place and is unclear how it is related to the topic. For example, "it is more efficient for agents..." What is efficiency exactly, and how is it compared across CounterDQN and MADQN?
* The reasoning and motivation for using positional differential games instead of Markov games as the framework are unclear. Markov games can be deterministic.
* The discretization of action space makes it unclear under what conditions the main theorem still holds.
* The effects of time discretization are unclear. How does it affect the estimation error or the convergence?

**Questions:**

* In Equ. 3, should it be $t_{m+1}$ instead of $\tau_{m+1}$?

---

> ### Author Response · Authors · 2023-11-16
> **Response**
>
> Dear Reviewer,
>
> Thank you very much for your attention to our paper and your remarks. Indeed, they address very important issues. We thought about most of them and tried to illuminate them in the paper. However, it may not be articulated enough. To fix this, below we go into detail on each of your remarks, comment on it and describe the corrections in the paper to make it better.
>
> ***In section 2, the authors could provide more insights into what is the difference between the standard formulation and differential game and why it is important to have a deterministic payoff.***
>
> You are correct that this is a crucial issue, and we paid attention to it in the third paragraph of the Introduction section dedicated to Markov games. Following your recommendation, we have also added the following comments regarding this at the beginning of section 2:
>
> *Recent studies consider RRL problems within the framework of zero-sum Markov games in pure or mixed policies. In the case of pure policies, it is known (e.g., paper-rock-scissors) that Markov games may not have a value (Nash equilibrium), which conceptually prevents the development of centralized learning algorithms based on shared q-functions. In the case of mixed policies, such formalization may also be inappropriate if, according to the problem statement (for example, in the case of developing expensive or safe control systems), it is required to seek robust policies guaranteeing a deterministic payoff value. In this section, we describe the framework of the positional differential games, which allows us, on the one hand, to consider the pure agents' policies and deterministic values of payoffs and, on the other hand, to obtain the fact of the existence of a value in a reasonably general case.*
>
> ***How the fact of the PDG is important to this paper and when the Isaacs's condition is met or not met.***
>
> Thank you for this question. We discussed the importance of the fact that the Markov game from Section 3 is a time-discretization of some differential game before Theorem 1. This is what helps us to prove it. We also tried to highlight the importance of the fulfilment and non-fulfilment of Isaacs's condition in the Limitations section. Following your recommendations, we also added the following short comments at the end of section 2:
>
> *We also note that in order to obtain further results, it is essential not only the existence of a value but also the fulfilment of Isaacs's condition as such.*
>
> ***In the experiment section, the metric of 'stability' appears without proper definition and explanation. For example, is 'stability' equivalent to 'deterministic payoff'?***
>
> Thank you for this comment. We mean stability with respect to running. That is, we say the algorithm is stable if it shows a similar result in all 5 runnings. In Figure 3, this means a narrow pale bar. The algorithm is unstable if, in some of the runnings, the algorithm performs worse than in the others. This corresponds to a wide pale bar. For example, MADDPG on Swimmer is unstable because, on at least one of the runnings, it showed a wide bar. We have added additional clarification to the article, chenging
>
> *Thus, looking at such a visualization, we can make conclusions about the stability*
>
> to
>
> *Thus, looking at such a visualization, we can make conclusions about the stability (with respect to running)*
>
> and
>
> *which reflects, on the one hand, the potential ability of MADDPG to find policies close to optimal, but, on the other hand, its instability.*
>
> to
>
> *which reflects, on the one hand, the potential ability of MADDPG to find policies close to optimal, but, on the other hand, its instability with respect to running.*
>
> ***In the experiment section again, some analysis seems to be out of place and is unclear how it is related to the topic.***
>
> Let us clarify how the experimental analysis relates to the topic of the paper. The main goal of the paper is to train policies that are robust with respect to the opponent's actions. In our experiments, we measure precisely this robustness. To do this, we consider a comprehensive evaluation scheme (the details of which can be found in the Experiments section) and offer a corresponding visualization.

---

> > ### Author Response · Authors · 2023-11-16
> > **Response**
> >
> > ***For example, "it is more efficient for agents..." What is efficiency exactly, and how is it compared across CounterDQN and MADQN?***
> >
> > Thank you for this question, which prompted us to insert additional explanations into the paper. By efficiency, we mean the exploitability of trained agents, that is, the difference between the obtained approximations of their guaranteed results $V_u^{\pi_u}$ and $V_v^{\pi_v}$ describing the distance to a value (Nash equilibrium). In Figure 3, this exploitability is expressed in bar width. The smaller the bar width, the better the algorithm efficiency. We have supplemented the paper with a brief explanation of this at the end of the Evaluation scheme paragraph:
> >
> > *The width of the bars illustrates the exploitability of both agents, that is, the difference between the obtained approximations of $V_u^{\pi_u}$ and $V_v^{\pi_v}$. If they are close to the optimal guaranteed results $V_u$  and $V_v$, then the width should be close to zero (if a value exists ($V_u = V_v$)). Thus, looking at such a visualization, we can make conclusions about the stability (with respect to running) and efficiency (with respect to exploitability) of the algorithms.*
> >
> > ***The reasoning and motivation for using positional differential games instead of Markov games as the framework are unclear. Markov games can be deterministic.***
> >
> > You are right that Markov games can be deterministic. We discussed why their consideration would not lead to a shared q-function in Section 3 before Theorem 1. Namely, arbitrary deterministic Markov games may have no value (or Nash equilibrium) (for example, in rock-paper-scissors). The feature of the class of deterministic Markov games that we are considering is that they are a time-discretization of differential games for which the Isaacs condition is satisfied. This allows us to obtain the result of Theorem 1 and derive effective algorithms. We also further covered this issue in the first paragraph of section 2.
> >
> > ***The discretization of action space makes it unclear under what conditions the main theorem still holds.***
> >
> > Thank you for your comment. We have added the following explanation in Corollary 1 (formerly Remark):
> >
> > *Let Isaacs's condition (6) holds. Let the value function $V(\tau,w)$ be continuously differentiable at every $(\tau,w) \in [0,T] \times \mathbb R^n$.*
> >
> > ***The effects of time discretization are unclear. How does it affect the estimation error or the convergence?***
> >
> > From the results of Theorem 1 (a), it can be shown that the smaller the time-discretization step $d(\Delta)$, the smaller the error $\varepsilon$. This can be proved by analogy with how the definition of the limit by sequences is obtained from $(\varepsilon,\delta)$-definition of the limit. A specific estimate, in our case, can be obtained from the proof of the theorem, but it is unlikely to be of practical interest since it will typically depend on the continuity property of the right-hand side of dynamical system (1) and quality index (2).
> >
> > ***In Equ. 3, should it be $t_{m+1}$ instead of $\tau_{m+1}$?***
> >
> > Thank you very much for this note. This is indeed a typo, which we have corrected in the updated version of the paper.
> >
> > Thank you again for your feedback, which allows us to add additional important clarifying comments to the paper. We hope these comments have improved the paper and our response has helped clear up some doubts. If this is the case in your opinion, then we respectfully ask that you consider increasing your score. If you have any more comments, questions or remarks, we would be happy to discuss them.

---

### Official Review · Reviewer_h5zf · 2023-11-02

**Soundness:** 3 good
**Presentation:** 3 good
**Contribution:** 3 good
**Rating:** 8
**Confidence:** 3

**Summary:**

Robust Reinforcement Learning (RRL) treats uncertainty as actions of an adversarial agent, and in this paper, the authors propose a novel approach by applying positional differential game theory to develop a centralized Q-learning method. The authors demonstrate that this method can approximate solutions to both minimax and maximin Bellman equations, and they introduce two algorithms, Isaacs Deep Q-Networks (IDQN) and Decomposed Isaacs Deep Q-Networks (DIDQN). The algorithms are tested in various environments and outperform other baseline RRL and Multi-Agent RL algorithms in the experiments.

**Strengths:**

The paper is well-organized and involving RRL in potential differential games is a neat idea. The ideas for the experiments all seem very natural to me. The paper is also nicely organized.

**Weaknesses:**

I think the authors address the limitations of their own work as is.

**Questions:**

Q1) In Section 5, when the environments is described, what is the difference between $R$ and $\mathbb{R}$?

Q2) What are the specifications of the system used to run the experiments?

Minor comments:

I think you should name your main theorem. At the moment, you referred to it as "Theorem" and I think it would help if you name it. Either Thereom 1, Main Theorem, or some other name. Similarly, I would enumerate the remark on page 5 and maybe call it a corollary.

I was unfamiliar with some of your notation, but if it is common in the field then you should keep it, of course. In particular, $\overline{a,b}$  to refer to the integers $a,a+1,\ldots, b$ was new to me. Similarly, I had to look up $\lim_{\delta\downarrow0}$. I merely wanted to point out that I wasn't familiar with the notation, though I guess it is not so hard to figure out from context.

---

> ### Author Response · Authors · 2023-11-16
> **Response**
>
> Dear Reviewer,
>
> Thank you very much for your positive feedback on our paper. We are glad you liked the paper in terms of organization and ideas. Also, thank you for your questions and comments. We are happy to answer them below.
>
> *Q1) In Section 5, when the environments is described, what is the difference between $R$ and $\mathbb R$.*
>
> Thank you very much for your note. It's a typo. There should be $\mathbb{R}^n$ everywhere. We corrected this in the updated version of the paper.
>
> *Q2) What are the specifications of the system used to run the experiments?*
>
> In our experiments, we use both well-known examples of differential games and examples based on the MuJoCo simulator. A brief description of them is given in the Environments paragraph of the Experiments section. A detailed mathematical description is given in Appendix E. If you have additional questions regarding it, we will be glad to answer them.
>
> *I think you should name your main theorem. At the moment, you referred to it as "Theorem" and I think it would help if you name it. Either Thereom 1, Main Theorem, or some other name. Similarly, I would enumerate the remark on page 5 and maybe call it a corollary.*
>
> Thank you for this comment. Indeed, it is better to call the theorem as "Theorem 1", and the remark as "Corollary 1".
>
> *I was unfamiliar with some of your notation, but if it is common in the field then you should keep it, of course. In particular, $\overline{a,b}$ to refer to the integers $a,a+1,\ldots,b$ was new to me. Similarly, I had to look up $\lim\limits_{\delta \downarrow 0}$. I merely wanted to point out that I wasn't familiar with the notation, though I guess it is not so hard to figure out from context.*
>
> Thank you for this note. Indeed, these notations are quite common in our specific field, but we are happy to clarify them so that they do not raise doubts among a wider circle of readers. Namely, we change the notation for consecutive integers $i \in \overline{a,b}$ to $i=a,a+1,\ldots,b$ and the notation the right limit at zero $\delta \downarrow 0$ to $\delta \to 0+$. The paper is only get better from this.
>
> Thank you again for your feedback. If you have any more comments, questions or remarks, we would be happy to discuss them.

---

> > ### Comment · Reviewer_h5zf · 2023-11-16
> >
> > Thank you for your comments.

---

### Official Review · Reviewer_o2s5 · 2023-11-07

**Soundness:** 1 poor
**Presentation:** 1 poor
**Contribution:** 2 fair
**Rating:** 5
**Confidence:** 3

**Summary:**

The paper discusses RRL, Robust reinforcement learning, a recent method to incorporate physics and other constraints into the RL paradigm, such as disturbances and perturbations. Finding algorithms for RRL (modeled as an extra adversary in an multi-agent RL setting) is difficult due to non-stationarity.

The paper introduces a shared-Q-function to compute policies, and compares their new algorithm (IDQN) to other algorithms, on a range of suitable games.

The contribution/text of the paper is mostly theoretical, proving theorems on why such an algorithm may work.

**Strengths:**

Robust RL is a new and under-studied problem, mostly due to the non-stationary state space. It is nice to see the problem being studied. Also nice is the part on the Isaac condition, and the derivation of the shared Q function algorithm. The experimental results comapring the performance to other algorithms is also nice

**Weaknesses:**

A shared-Q function algorithm is compared against non-shared Q MARL algorihtms, or to standard single agent RL algorithms such as DDQN and PPO. This is comparing apples and oranges. Of course  shared Q function algorithms outperforms the other options. As such the experimental results are not very meaningful.
Sharing the Q function is not really addressing the MARL problem of non-stationarity, it is a kind of cheating.

**Questions:**

I find the paper sympathetic in that it addresses Robust RL. Using a shared Q function transforms the MARL problem into a single agent problem. The pages of proofs do not contribute much to deeper understanding of the problem.

---

> ### Author Response · Authors · 2023-11-16
> **Response**
>
> Dear Reviewer,
>
> Thank you very much for your attention to our paper, as well as for your comments. It is very nice to hear positive feedback regarding Isaacs's condition, the derivation of the shared q-function, and the experimental part. Below, we present our comments on your remarks in the Weaknesses and Questions sections.
>
> ***A shared-Q function algorithm is compared against non-shared Q MARL algorihtms, or to standard single agent RL algorithms such as DDQN and PPO. This is comparing apples and oranges. Of course shared Q function algorithms outperforms the other options. As such the experimental results are not very meaningful.***
>
> Thank you for this remark. We understand your doubts regarding the choice of the baseline algorithms, and therefore, we would like to present the following arguments in defence of our particular choice of them.
>
> The decentralized approach is widely studied in general MARL problems and is sometimes quite successful. Therefore, it was essential for us to show that this approach does not work well for the zero-sum MARL under consideration. The results of the DDQN algorithm as an example of the decentralized approach show that this is indeed true. The comparison with the RARL algorithm is important because it represents the first approach proposed in the literature (Pinto et al. (2017)) for solving Robust RL problems. Comparison with the non-shared MARL algorithms (MADDPG and MADQN) is a kind of ablation. We verify experimentally that the shared q-function actually increases performance. In the general case, this fact does not seem obvious since even in our experiments, non-shared MADQN outperforms NashDQN with a shared q-function. Thus, the baseline algorithms chosen for comparison help us draw several important conclusions from the experiments, a summary of which is presented in the last paragraph of the Experiments section.
>
> ***Sharing the Q function is not really addressing the MARL problem of non-stationarity, it is a kind of cheating.***
>
> If we understand you correctly, you doubt that the shared q-function can be used to solve general MARL problems. If this is so, we are generally ready to agree since the general formulation of MARL may concern MARL with more than two agents pursuing not necessarily opposite goals. However, this paper focuses on two-agent competitive MARL problems that can be described as zero-sum differential games. This is a limitation of the paper, which nevertheless allows us to obtain theoretical results and effective algorithms (IDQN and DIDQN) superior to the baseline algorithms aimed at solving more general MARL (MADDPG and MADQN). Some considerations about when this limitation can be overcome are indicated in the Limitations section. If you meant something else, please feel free to clarify.
>
> ***I find the paper sympathetic in that it addresses Robust RL. Using a shared Q function transforms the MARL problem into a single agent problem. The pages of proofs do not contribute much to deeper understanding of the problem.***
>
> It's great to hear that you find the paper sympathetic. Let us explain why it may not be entirely correct to say that we are reducing the problem to a single agent. One agent pursues maximizing its rewards, whereas in our case, we simultaneously train two agents who pursue opposing goals. This approach (centralized approach) is reasonably typical for solving MARL problems and, in our paper, we develop it for a specific class of MARL. Do not hesitate to ask if you have additional questions regarding a deeper understanding of the problem.
>
> Thank you again for your feedback. We hope that our response has helped to clear up some doubts. If this is the case in your opinion, then we respectfully ask that you consider increasing your score. If you have any more comments, questions or remarks, we would be happy to discuss them.

---

> > ### Comment · Reviewer_o2s5 · 2023-11-22
> >
> > Thanks very much for the detailed answers. I understand your point, but remain unconvinced about the shared Q approach. I believe my original score is accurate (perhaps somewhat generous)

---

> > > ### Author Response · Authors · 2023-11-23
> > > **Response**
> > >
> > > Thank you very much for your feedback

---

### Author Response · Authors · 2023-11-16
**Сhanges in the paper**

Many thanks to all reviewers for their valuable comments. Based on them, we made the following changes to the paper, which, as we hope, made it more motivating and clear:

1. We added the following comments at the beginning of the Positional Differential Game section to explain the motivation for using differential games instead of Markov games in pure or mixed policies:

*Recent studies consider RRL problems within the framework of zero-sum Markov games in pure or mixed policies. In the case of pure policies, it is known (e.g., paper-rock-scissors) that Markov games may not have a value (Nash equilibrium), which conceptually prevents the development of centralized learning algorithms based on shared q-functions. In the case of mixed policies, such formalization may also be inappropriate if, according to the problem statement (for example, in the case of developing expensive or safe control systems), it is required to seek robust policies guaranteeing a deterministic payoff value. In this section, we describe the framework of the positional differential games, which allows us, on the one hand, to consider the pure agents' policies and deterministic values of payoffs and, on the other hand, to obtain the fact of the existence of a value in a reasonably general case.*

2. We added the following short remark at the end of the Positional Differential Game section regarding the importance of fulfilling Isaacs’s condition for the paper results:

*We also note that in order to obtain further results, it is essential not only the existence of a value but also the fulfilment of Isaacs's condition as such.*

3. We changed "Theorem" to "Theorem 1" and "Remark" to "Corollary 1".

4. We added to the beginning of Corollary 1:

*Let Isaacs's condition (6) holds. Let the value function $V(\tau,w)$ be continuously differentiable at every $(\tau,w) \in [0,T] \times \mathbb R^n$.*

5. We changed the notation for consecutive integers $i \in \overline{a,b}$ to $i=a,a+1,\ldots,b$ and the notation the right limit at zero $\delta \downarrow 0$ to $\delta \to 0+$.

6. We explained the stability of the algorithms in the following sentences:

*Thus, looking at such a visualization, we can make conclusions about the stability (with respect to running) and efficiency (with respect to exploitability) of the algorithms.*

*Regarding average by runnings, the algorithm is also well in HomicidalChauffeur, InvertedPendulum, and Swimmer, which reflects, on the one hand, the potential ability of MADDPG to find policies close to optimal, but, on the other hand, its instability with respect to running.*

7. We further explained at the end of the Evaluation Scheme paragraph how the bar width describes the efficiency of the trained policies:

*The width of the bars illustrates the exploitability of both agents, that is, the difference between the obtained approximations of $V_u^{\pi_u}$ and $V_v^{\pi_v}$. If they are close to the optimal guaranteed results $V_u$  and $V_v$, then the width should be close to zero (if a value exists ($V_u = V_v$)). Thus, looking at such a visualization, we can make conclusions about the stability (with respect to running) and efficiency (with respect to exploitability) of the algorithms.*

8. We also fixed typos by changing $R$ to $\mathbb R$ and $\tau_{m+1}$ to $t_{m+1}$.

---

### Meta-Review · Area_Chair_CcGr · 2023-12-09

**Metareview:**

This submission has undergone extensive review, generating a mix of opinions from the reviewers and the area chair. The primary focus of the paper is on leveraging zero-sum positional differential games to enhance robustness in reinforcement learning through the development of centralized Q-learning methods. The authors propose two algorithms, Isaacs Deep Q-Networks (IDQN) and Decomposed Isaacs Deep Q-Networks (DIDQN), and compare their performance with existing reinforcement learning and multi-agent RL algorithms in various environments.

## Areas of Strength:
- The concept of applying positional differential game theory to robust reinforcement learning (RRL) is recognized as novel.
- The submission demonstrates a deep theoretical understanding of the subject.
- Reviewer h5zf noted the paper’s good organization and clarity in presentation.

## Areas of Concern:
- Reviewer BWN5 pointed out issues with the paper’s writing quality and organizational structure. Specifically, a lack of smooth transitions between sections and unclear motivations for using positional differential games.
- The same reviewer highlighted that even typos from a previous version of the paper were not corrected, raising concerns about the authors’ attention to detail.
- Concerns about the experimental setup and comparisons were raised, questioning the effectiveness of the shared Q-function approach. Reviewer o2s5 remarked that comparing a shared-Q function algorithm against non-shared Q MARL algorithms is akin to comparing apples and oranges.
- The authors’ responses to the reviews, while comprehensive, did not fully convince some reviewers, such as o2s5 and BWN5, of the merits of the approach or the adequacy of the revisions.

**Justification For Why Not Higher Score:**

In light of these reviews, the recommendation is to reject this submission. The key reasons include concerns about the clarity and organization of the paper, the unaddressed errors from previous versions, and doubts about the experimental methodology’s efficacy. The novelty of applying positional differential game theory in RRL, while recognized, is overshadowed by these issues. Future submissions would benefit from a more detailed justification of the chosen approach, better attention to writing quality, and a more convincing experimental setup.

**Justification For Why Not Lower Score:**

N/A

---

### Decision · Program_Chairs · 2024-01-16

Reject